# Online Dynamic Network Visualization Based on SIPA Layout Algorithm

**Guijuan Wang** [1,†], **Huarong Chen** [1,†], **Rui Zhou** [1], **Yadong Wu** [2,*], **Wei Gao** [1], **Jing Liao** [1] and **Fupan Wang** [1]

1   School of Information Technology and Computer Science, Southwest University of Science and Technology, Mianyang 621010, China; guijuanwang@swust.edu.cn (G.W.); chenhuarong@swust.edu.cn (H.C.); gaowei_2010@126.com (W.G.); liaojing@swust.edu.cn (J.L.)
2   School of Computer Science and Engineering, Sichuan University of Science and Technology, Zigong 643002, China
*   Correspondence: wyd028@163.com
†   These authors contributed equally to this work.

**Abstract:** Online dynamic network visualization is imperative for real-time network monitoring and analysis applications. It presents a significant research challenge for maintaining both layout stability and quality amid unpredictable temporal changes. While node-link diagrams are extensively utilized in online dynamic network visualization, previous node-link-diagram-based research primarily focused on stabilizing the layout by defining constraints on local node movement. However, these constraints often neglect the structural influence and its corresponding global impact, which may lead to that the representations of the network structure change significantly over time and a decrease in layout quality. To address this problem, we introduce the Structure-based Influence Propagation and Aging (SIPA) algorithm, a novel approach to preserve the stability of relative node positions and shapes of interconnected nodes (referred to as structures) between adjacent time steps. These stable structures serve as visual cues for users tracking the evolution of the network, thereby enhancing the overall layout stability. Additionally, we enhance dynamic network analysis by a highly interactive visualization system, enriching the layout result with multiple coordinated views of temporal trends, network features, animated graph diaries and snapshots. Our approach empowers users to interactively track and compare network evolution within a long-term temporal context and across multiple aspects. We demonstrate the effectiveness and performance of our approach through in-lab user studies and comparative experiments with three baseline dynamic network layout methods.

**Keywords:** dynamic network; graph layout; online; mental map; visualization





## 1. Introduction

A network is the combination of a set of entities and the relationships between them [1]. Network visualization provides a visual means for analyzing the relation data and finds wide-ranging applications across multiple domains [2], spanning from transportation systems [3], social networks [4], biological networks [5,6], contact tracings [7], to academic writing analysis [8]. By nature, a significant portion of these networks is dynamic, where nodes and edges evolve over time. The challenge in visualizing dynamic networks is to compute a new layout that are both aesthetically pleasing and fits well into the sequence of drawings of the evolving graph [9]. The former criterion is about layout quality [10], and the latter criterion is about maintaining the layout stability, also termed as the mental map [11].

Many dynamic layout methods have been developed to address these challenges. The offline methods, where the entire sequence of the graphs to be drawn is known, can pre-adjust node positions to accommodate forthcoming graph changes [12,13] and produce relatively stable layouts across the entire time spans. In contrast, for real-time network monitoring and analysis applications, online layout methods are required. Since

the upcoming changes are unknown, online layout methods are more challenging in terms of stability maintenance [9,14]. Node-link diagrams are extremely common and is a widely employed network layout technique [15]. Existing node-link-diagram-based online layout algorithms [9,16,17] primarily address this challenge by restricting the movements of local nodes while overlook broader influences, potentially resulting in failures to adequately represent the stable structures or the overall shape of the network. In this context, the network structures refer to the shape and relative positions for interconnected nodes. Stable structures persisting across consecutive time steps serve as anchors, facilitating users in tracing the network evolution, and therefore are important for ensuring the overall layout stability. Consequently, the development of new techniques becomes imperative to augment the handling of this issue. On the one hand, enhancing layout techniques is essential. On the other hand, it is challenging for human users to fully comprehend the network evolution by solely viewing the layout result. Visualization systems with multiple auxiliary views and rich interactions can effectively enhance user perception.

To address these challenges, we introduce a novel Structure-based Influence Propagation and Aging (SIPA) dynamic node-link diagram layout algorithm, and augmente the layout result with the design of an interactive visualization system. This not only enhances users' understanding of evolving network structures, but also facilitates the exploration and comparison of network dynamics. Our main contributions in this research can be summarized as follows:

1.  We extend previous online dynamic layout methods by a novel SIPA layout algorithm. This algorithm is proposed based on the influence of structural changes to different nodes, and with a combination of node ages. While ensuring layout quality, our algorithm better preserves the relative positions and shapes of structures that persist across adjacent time steps. These stable structures provide anchors for tracing the network evolution and thus contributes to enhancing the overall layout stability.
2.  We design and implement an interactive visualization system that enriches dynamic network analysis with multiple coordinated views. The system provides crucial temporal aspects and features of dynamic networks, enhancing exploration, tracking, and comparison of network dynamics.
3.  We verify the performance of our algorithm by comparative experiments based on three dynamic network datasets; and we demonstrate the usability and effectiveness of our system through use cases and a user study.

The following sections of this paper is organized as follows: we first review the related work in Section 2. We then present our dynamic network layout algorithm in Section 3 and the visualization system in Section 4. Section 5 describes the evaluation process through comparative experiments on real-world datasets, use cases and user study. We conclude our work in Section 7. Through this research, we aim to contribute to the advancement of online dynamic network visualization for researchers and practitioners across various domains.

## 2. Related Work

In this section, we review two research areas that are closely related to our work: the dynamic network layout methods and the dynamic network visualization techniques.

### 2.1. Dynamic Graph Layout Methods

Many algorithms have been proposed on dynamic network layout. These algorithms aim to preserve user's mental maps [11,18,19] as network structures change over time, ensuring a coherent layout sequence that aids users in understanding the network evolution, yet the layout quality must adhere to aesthetic criteria [10,20]. Based on data requirement, dynamic network layout methods can be classifieds into two categories: offline methods and online methods.

**Offline method:** Offline methods, where all network states of the entire time spans are known, can pre-adjust node positions to accommodate forthcoming nodes, and minimize layout changes to better maintain the user's mental map, often employing the concept of

"hypergraphs [12]" for layout. However, such offline methods have higher demand on data, all network states to be visualized need to be known in advance, and does not suit to the real-time dynamic network layout scenarios.

**Online method:** Many real-world networks are also likely to be streaming over time. A streaming graph is a continuous, unbounded, rapid, time-varying stream of edges that is clearly too large to fit in memory except for probably short windows of time [21]. For monitoring the streaming networks in real time, online dynamic layout methods are required. Online layout methods face greater challenges considering the upcoming changes are unpredictable. Their goal is to minimize visual changes between the original and modified layouts when new network update comes, enabling users to swiftly comprehend graphical changes. Lin [19] was among the pioneers in online layout algorithms, developing techniques that conserve mental maps while generating aesthetically pleasing layouts. Common practice in online network drawing involves adding constraints to nodes, limiting their mobility and ensuring a consistent mental map across the sequence of graphs [22]. Pinning algorithms [22] employ simulated annealing to reduce node displacement between adjacent layouts. The Aging algorithm [16] introduces node and edge age, calculated based on node appearance time and lifetime exposure to motion, yielding relatively stable positions for long-term stable nodes. Crnovrsanin et al. [17] extended FM3 to accommodate dynamic network data, proposing a refinement scheme grounded in node energy to allow highly influential nodes to adjust their positions within the layout. Dwyer et al. [23] introduced the DIG-COLA algorithm, building on the KK algorithm and incorporating directional constraints for drawing directed networks. Yuan et al. [24] and Wang [25] introduced multiple constraints, encompassing circular, distance, and directional constraints, even enabling users to define structures to generate network layouts. LCDE [26] employs a distance-constrained stress model in online dynamic network drawing. These methods primarily handle the local changes, while neglect the influence to the global structure and shapes.

In this paper, we contribute to the online dynamic node-link diagram layout field by introducing the Structure-based Influence Propagation and Aging (SIPA) layout algorithm. The SIPA is tailored to address the challenges of balancing the network stability and overall network layout quality.

### 2.2. Dynamic Network Visualization Approaches

Dynamic network visualization have been a hot research topic for years that tackles the challenge of representing the evolution of relationships between entities in comprehensible, scalable, and effective diagrams [12].

First, based on the representation of the temporal dimension, the visualization techniques can be categorized as animation-based, timeline-based, and hybrid. **Animation-based techniques** [27–29] use animations to depict changes between adjacent network layout sequences, facilitating users in observing and understanding the trends in network changes. However, due to the challenge of users remembering previous network states, animation-based methods may not support effective comparisons between network structures at different time steps. **Timeline-based techniques** [6,30] replace animation with static views, allowing users to analyze network structure details. Nevertheless, when dealing with large scale network sequences, fully representing them within limited screen space can be challenging [12]. **Hybrid techniques** combine the animation and timeline methods for time representation, leveraging each one's advantages, such as the Flip-Book [31] presented the long network sequence with a moving time line; MultiPiles [32] designed small multiple views of piling adjacency matrices that can be animated, offering the ability to scale to networks with hundreds of temporal snapshots. The comparative study on animation-based and timeline-based shows that sequences of timeline-based static methods perform better on precision tracking [16], and animations are suitable for present broad high-level changes [33]. Therefore, we adopt a hybrid method, using the animated layout

to show the overview, and providing small multiple network snapshot views for further accurate analysis and comparisons.

Second, network representation technique is another important aspect for dynamic network visualization. This topic have be extensively studied in static and dynamic network visualizations. Comprehensive reviews can be found in [15,34,35]. Typical network visualization forms are node-link diagrams, matrix diagrams and space-filling diagrams. The space filling techniques, such as treemap or sunburst diagrams, have advantages on screen space utilization. This technique is suitable to hierarchical networks for the space explicitly encode hierarchical relations [15]. Comparative study on node-link diagrams versus matrices diagrams shows that the matrices perform better on larger network (100 vertices), while the node-link diagrams are more familiar to users and more readable in small size network (20 vertices) [36]. Besides the node-link diagram are widely employed and can better present the topological structures [15]. For dynamic network visualization, the time can also be an encoding dimension, the Massive Sequence View [37–39] is a timeline-based technique for dynamic network visualization. The node positions are consistent over time, and no node clutter problems, so it has demonstrated to be a more scalable solution [38]. Considering our layout algorithm is built on node-link diagram. We select the node-link diagram for our visualization system design.

Furthermore, the dynamic network exploration is generally enhanced by multiple interactively coordinated views [40], combinations of small multiples [13], or the Level Of Details (LOD) techniques [41].

These studies are formative for our approach. Our visualization system further supports users' exploration of dynamic network by a hybrid technique, rich interactions, and multiple linked temporal and featural views.

## 3. Dynamic Network Layout Algorithm

Given a dynamic network $N = \{N_0, N_1, \ldots, N_n\}$, our SIPA algorithm aims to continuously online compute the corresponding layout sequence $L = \{L_0, L_1, \ldots, L_n\}$, ensuring the overall layout quality while maintaining layout stability between adjacent networks. When the network transitions from $N_{i-1}$ to $N_i$, the network updates $U_i$ may involve the addition or removal of nodes and edges. For the unchanged portions of $N_{i-1}$, the node positions are directly copied from the layout $L_{i-1}$. Figure 1 depicts the workflow of our algorithm. The input of our algorithm is a streaming graphs. The objective of the SIPA algorithm is continuously compute node-link-diagram-based layouts when changes of the upcoming graph is unknown, and try to preserver the layout stability while ensuing the layout quality. The main steps of SIPA dynamic network layout algorithm consist of: (1) initial placement of newly added nodes; (2) determining structural influences on nodes and propagating influences within the network; (3) combining the node influence with node aging strategy to further preserving the stability; and (4) updating layout with combination of the former factors.

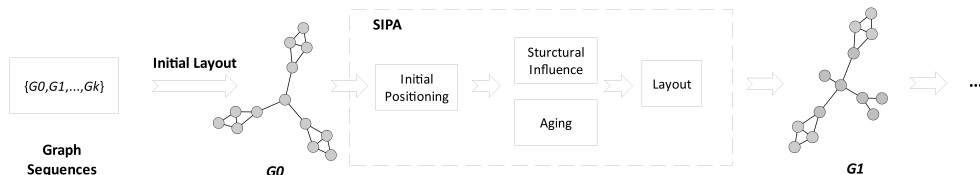

**Figure 1.** The flow diagram of the SIPA algorithm.

### 3.1. Initial Positioning for Newly Added Nodes

The position of a node is generally determined by the collective decisions of all its neighboring nodes. When the positions of a node's neighboring nodes are already established, ideally, the node should be placed at the centroid or weighted centroid of all its neighboring nodes, as shown in Figure 2. However, this approach has some drawbacks. First, when adding not just an isolated node but a set of nodes which forms an connected

structure, simultaneously adding these nodes result in unreasonable initial positions for the newly structure. Moreover, when the added nodes share the same neighboring nodes, these nodes will have the same initial positions, causing node overlap.

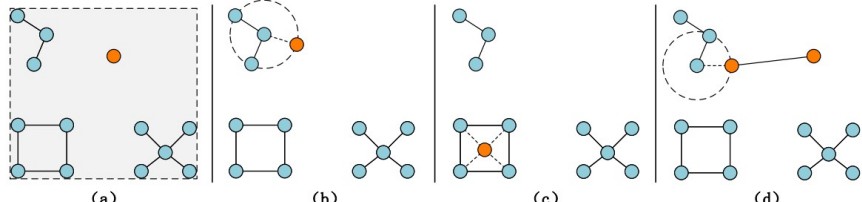

**Figure 2.** Centroid-based initial positioning for new nodes. A new node may: (**a**) be a isolated node, (**b**) have one edge with a existing node, (**c**) have edges with multiple existing nodes, and (**d**) have one edge with a newly added node.

To address these issues, we employ the Sorted Sequential Barycenter Merging (SSBM) method [42] to assign positions to the newly added nodes. The positions of the nodes to be added are determined sequentially based on the number of connections with existing nodes, thereby reducing the probability of random placement of nodes, as shown in Figure 3.

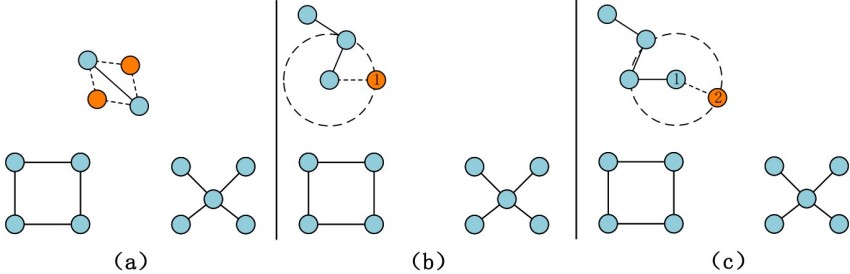

**Figure 3.** SSBM-based initial positioning for new nodes. The new node may: (**a**) have edges with multiple existing nodes, (**b**) have one edge with a existing node, and (**c**) have one edge with a newly added node.

The calculation of positions for the newly added nodes is as Formula (1).

$$
x_i = \begin{cases} \frac{1}{|C_i|} \sum_{j \in C_i} x_j + e_1, & |C_i| \geq 2 \\ x_j + e_2, & |C_i| = 1 \text{ and } j \in C_i \\ e_3 & |C_i| = 0 \end{cases} \tag{1}
$$

where $C_i$ represents the set of nodes that are already assigned positions and are connected to the new node $n_i$. $x_i$ represents the position coordinates of node $n_i$, and $x_j$ represents the position coordinates of node $n_j$. $e_1$, $e_2$ and $e_3$ are small random vectors that are used to ensure initial positions are not the same for nodes sharing identical neighboring nodes.

### 3.2. Structure-Based Influence Computation and Propagation

We introduce a novel node movement constraint strategy based on the factor that nodes are influenced by structural changes. When the network transitions from state $N_{i-1}$ to $N_i$, the network changes directly affect one or more nodes within the network, requiring adjustments to their positions due to alterations in their connectivity. We quantify the network structural influence factor on different nodes and propagate this influence factor through the network using a depth-first-search (DFS) graph traversal approach. This allows more nodes to have the opportunity to readjust their positions, ultimately leading to the attainment of an ideal layout.

(1) Node Influence Calculation

In previous methods, affected nodes were typically labeled as movable nodes without considering the varying degrees of impact different changes have on different nodes. As illustrated in Figure 4, when removing an edge from the network (indicated by dashed lines in the figure), it affects two vertices connected by this edge, denoted as nodes $u$ and $v$, rendering their existing positions no longer suitable, thus necessitating a recalculation of their positions. Following the removal of this edge, the degree of node $v$ becomes 1, meaning it has only one neighboring node left, whereas the degree of node u remains 8. The re-positioned nodes should appear as depicted in Figure 4. Comparatively, the position change of node $v$ is much greater than the change of node $u$. This discrepancy arises because a node's position is determined collectively by its neighboring nodes. The node with the higher degree should experience a smaller impact than the node with the lower degree.

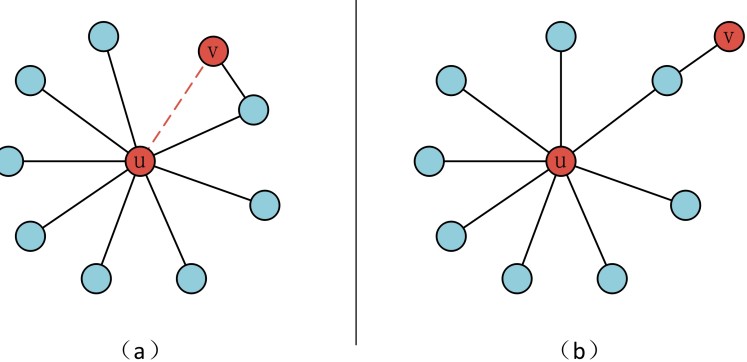

**Figure 4.** The influence of structural changes on different nodes: (**a**) before the change, (**b**) after the change.

Hence, in this study, when calculating the influence of network structural changes on different nodes, the structural information of nodes is taken into account. The influence value $I_i$ on node $i$ ($n_i$) are computed based on the degree information before and after the changes as Formula (2).

$$I_i(t) = \begin{cases} min(\frac{d_i^{add}(t)+d_i^{del}(t)}{d_i(t-1)}, 1) & n_i \in N(t-1) \\ 1 & n_i \notin N(t-1) \end{cases} \tag{2}$$

where $d_i^{add}(t)$ and $d_i^{del}(t)$ represent the deletion and addition degrees of $n_i$:

$$d_i^{del}(t) = |j \in N_i(t-1) \backslash N_i(t)| \tag{3}$$

$$d_i^{add}(t) = |j \in N_i(t) \backslash N_i(t-1)| \tag{4}$$

(2) Influence propagation

Nodes within the network are not isolated. The movement of one node will result in changes to all nodes connected to it. Similarly, localized changes within the network not only lead to alterations in their neighborhood structure but also impact the overall network structure. The influence degree that a node's movement has on its neighboring node is determined by the neighbor's degree. As illustrated in Figure 5, for node $c$, its movement affects node $d$ with a degree of 1 because node $d$ is only connected to node $c$. Conversely, the movement of $d$ has an impact of only $\frac{1}{3}$ on node $c$ because $c$ has three connections. Therefore, when node $u$ moves, the calculation of the influence degree $I_{uv}$ on node $v$ is as

Formula (5), where $d_v$ represents the degree of node v. In the case of an weighted network, the degree can be replaced with edge weights.

$$I_{uv} = \begin{cases} \frac{1}{d_v} & uv \in E \\ 0 & uv \notin E \end{cases} \tag{5}$$

Based on Equation (5), the propagation matrix of influence between nodes in an unweighted network can be calculated, as illustrated in Figure 5.

Furthermore, the propagation of influence values can be calculated based on the propagation matrix and the DFS. The influence propagates from directly affected nodes to the rest of the network, or to nodes of its connected component in the case of that the network is not a connected graph. As depicted in Figure 6, let's denote node 17 as the node that has been moved. Using the propagation matrix, one can calculate its influence values on neighboring nodes. Subsequently, this method is applied recursively based on the DFS to calculate the influence values of the remaining nodes within the network.

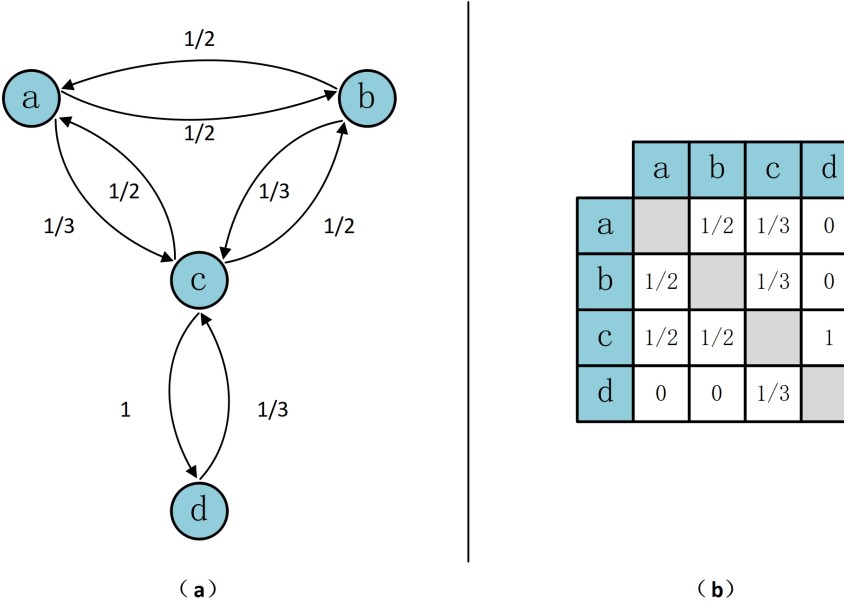

**Figure 5.** The propagation matrix (**b**) for the network topology (**a**).

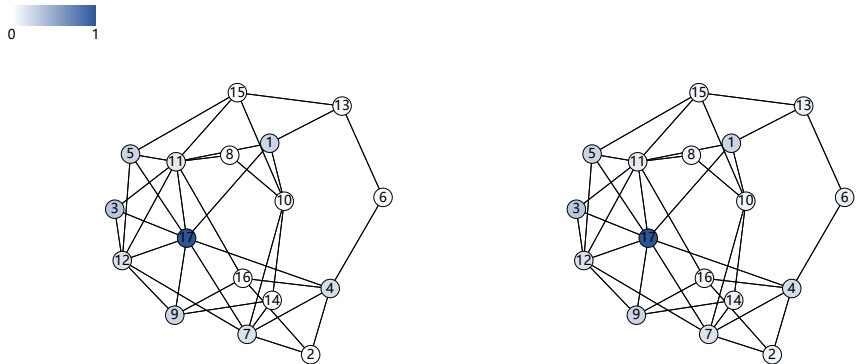

**Figure 6.** The influence propagation due to the movement of Node-17. (**Left**): influence generated on direct neighbours; (**Right**): influence has been propagated to indirect neighbours. Numbers in the circle represent node ID and colors of the circle encode the influence value.

### 3.3. Node Aging Strategy

We introduce the concept of "age" [16] to further stabilize the network layout and maintain user's mental map, by ensuring that nodes in the network that have remained unchanged for a long time are more stable. With this concept, the update process of dynamic network is considered as an aging process, where the age of nodes in the network is defined as the number of network updates that a node has experienced. Nodes with older ages are less mobile.

However, age constraints prevent old nodes that have experienced direct changes from reaching a reasonable location. Therefore, we define the age calculation strategy through categorization. Nodes that have never directly encountered changes are kept stable, while nodes that have directly encountered changes are encouraged to move in order to reach more reasonable positions. Let $a_i(t)$ represents the age of node $i$ at time $t$, and its update rules are as shown in Equation (6).

$$a_i(t) = \begin{cases} 1 & N(i) > 0 \\ a_i(t-1) + 1 & otherwise \end{cases} \tag{6}$$

Here, $N_i$ is the set of adjacent nodes to node $i$ that are updated at time $t$. If a node in the network has directly encountered changes, its age is reset to 1; otherwise its age is simply incremented by 1.

### 3.4. Node Mobility Factor and Layout

Finally, we define mobility factors for nodes by combining the influence and age factors. These mobility factors are used to adjust the forces acting on the nodes, ultimately yielding the final layout result.

The node's mobility in the subsequent layout is together determined by the node influence factor and node age. The larger node influence factor encourages nodes to move to achieve a better layout, while the larger node age stabilizes nodes in the network that haven't experienced updates for a long time. Consequently, a node's mobility in the network can be defined as Formula (7).

$$M_i(t) = \alpha \times I_i(t) + (1-\alpha) \times e^{-\beta a_i(t)} \tag{7}$$

where, the left portion represents the influence factor, the right portion corresponds to the age factor, and $\alpha \in [0,1]$ is the parameter for tuning the trade-off between network stability and the layout quality. The aging formula reuse the exponential decay approach in reference [16] and its $\beta \in IR$ is the aging rate which allows for tuning the evolutionary process. After combining the results of these two factors, the mobility factor is scaled within the range of $[0,1]$. The lower a node's mobility factor, the less likely it is to move during the layout process.

The mobility factor is then applied to the layout method to yield the final layout result, as a mean to alter the force on nodes to improve the layout stability. In our experiments, the value of $\alpha$ is set to 0.5. Users can adjust these tuning parameters from the user interface of the visualization system as needed, considering their preferences for layout quality, stability, and specific data characteristics.

## 4. Interactive Visualization System

The dynamic nature of network structures makes it challenging for users to fully comprehend the evolving trends in network structure solely based on the network layout view. To address this issue, we design a visualization system. The system pipeline is shown in Figure 7. The data include node data, edge data and temporal data. We first process the original data into network data, and compute network layout using SIPA; then the feature statistics such as degrees, communities, and temporal edge and node size are

calculated; Lastly the layout and the feature statistics are visualized in multiple views, and the animation and linking animation are adopted to further users progressive analysis.

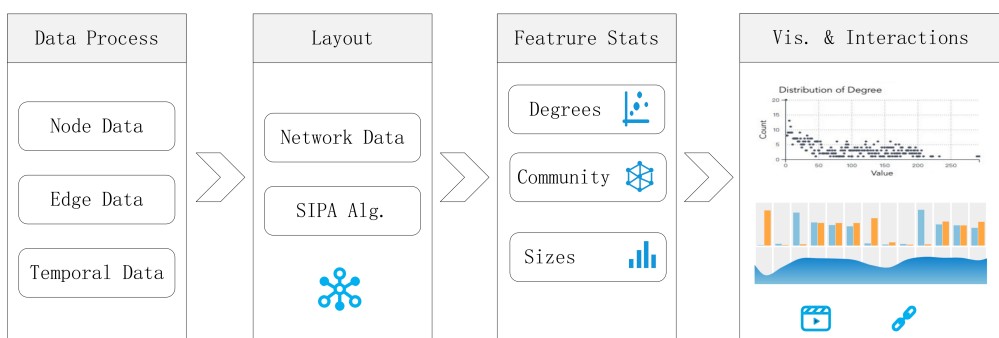

**Figure 7.** The pipeline of our dynamic network visualization system.

For the user interface design, to facilitate users to progressively understand the dynamic network evolvement, we adopted a "overview + details" design style. The animation-based technique provide a clean view on the broad high-level changes, while sequences of static network layouts are more capable of precision changes tracking [16]. So we arrange the animated layout in the main view to show the overall network status, and place the small-multiple-based static network snapshots besides the main view to support detailed analysis and comparisons. The user interface of our visualization system is depicted in Figure 8, comprising the control panel module (A), the main view module (B), and the network snapshot view module (C), and the temporal and featural views (D and E).

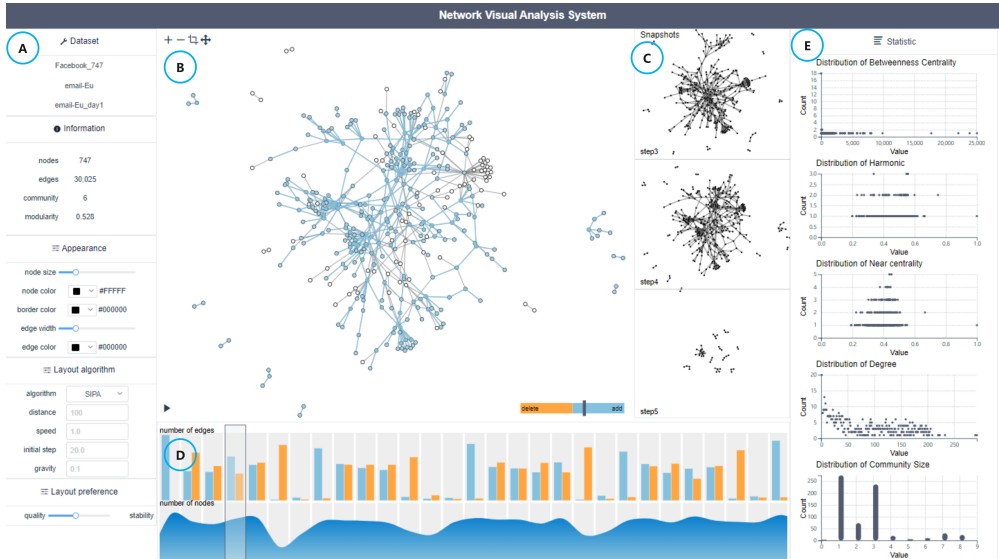

**Figure 8.** The user interface of dynamic network visualization system, which consists of (**A**) the control panel, (**B**) the main view, (**C**) the snapshots view, (**D**) the temporal views, and (**E**) the feature statistics view.

### 4.1. Visualization Design

Control Panel: This panel provides functions related to network dataset selection, visual appearance adjustments, network layout algorithm selection, and layout perference setting. The network dataset selection function enables switching or uploading network datasets, as the input for the entire system. Visual appearance control allows users to adjust visual properties of nodes and edges within the main view, including node radius size, node colors, edge width, and edge colors. Network layout algorithm selection enables switching among different layout algorithms, providing the system with the flexibility to support various algorithms. This offers users more choices and enhances system compatibility

with a wider range of data types. The SIPA algorithm is selected by default. If the layout result is not satisfactory, users can switch to FR, Incremental or Aging algorithms. The Layout preference panel allow users to set their preference on layout stability and quality. The slider is in the middle by default, user can move it based on their preference and the network layout results.

Main View: This view present the layout results of the network, which convey the topological structural information of the network. When there are significant structural changes in the network, such as the simultaneous addition or removal of multiple nodes and edges, the positions of network nodes undergo substantial alterations, making it difficult for users to comprehend how the network transitions from one structure to another. To address this, we employ a segmented animation approach inspired by the design of GraphDiaries [29]. The animation is implemented with the transition module of D3 [43] visualization library, The duration of a transition is set to 750 milliseconds for now, and future study needs to be performed for choosing a optimal duration value. The network's structural changes are divided into three stages: node deletion, node addition, and node movement, and highlighting is applied before structure deletion and after structure addition to emphasize the changes in network structure (see Figure 9). Additionally, during the node movement stage, to facilitate user observation and tracking of node position changes, we employ node movement interpolation and retain movement traces to better present the process of node position changes.

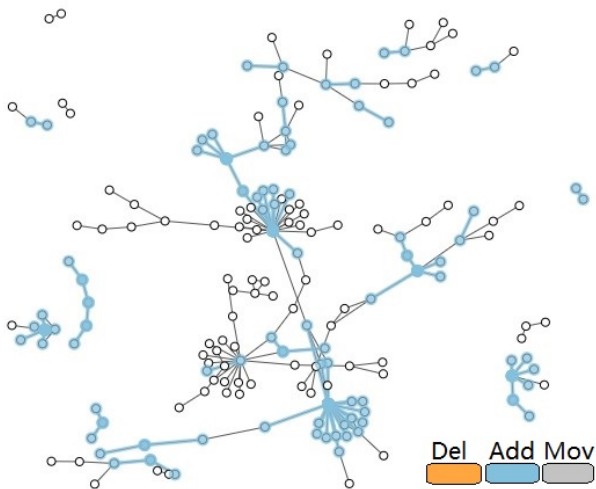

**Figure 9.** Highlighting network changes through 3-step (delete/add/move) animations.

Network snapshot views: Utilizing animation to represent dynamic networks aids users in observing and comprehending changes in network structure. However, due to the inherent volatility of dynamic network structures and limitations in users' memory capacity, it becomes challenging for users to simultaneously understand the current network structural changes while retaining information about the network's former structures. This impedes users' ability to compare structural differences between different time steps effectively and hinders their ability to gain an overview of the network's structural characteristics at various moments, as well as to identify patterns in network evolution. To address this issue, a Network Structural Snapshot module has been designed to store structural information from previous network states.

Network feature Statistics views: The networks itself possesses many characteristic features, such as the number of network nodes, edge count, degree information of nodes in the network, and the structural composition of network communities. While these information are valuable for understanding networks, expressing them through the layout diagram of the network is challenging. To facilitate users in rapidly comprehending the statistical feature information of the network, our system introduces a network feature statistics mod-

ule. This module presents node degree distribution, community quantity distribution, node betweenness centrality distribution, node closeness centrality distribution, and eccentricity distribution. Degree distribution reflects information about the degrees of nodes within the network and serves as an indicator of network type or structural features. Communities in a network are the dense groups of the vertices, which are tightly coupled to each other inside the group and loosely coupled to the rest of the vertices in the network [44]. The distribution of community quantities reflects both the number of communities in the network and the distribution of node quantities within each community. In our study, we employ the Louvain [45] community detection algorithm to identify communities within the network. Statistics of community distribution aids users in understanding the network. Additionally, users can use this information to estimate whether the layout in the main view reflects the clustering patterns within the network. Betweenness centrality, viewed from the perspective of nodes as "intermediaries", measures their importance and is defined as the count of the shortest paths passing through a particular node. From the community structure viewpoint, nodes with high betweenness centrality often play a critical role in bridging two communities. The distribution of node betweenness centrality reflects the distribution of betweenness centrality among nodes within the network and is a significant representation of network structural features. Closeness centrality of nodes reflects their distance from the network center. If all nodes in a network have low closeness centrality, it indicates a small network diameter.

Temporal views: The temporal views depict the long-term trend of dynamic networks. Given the stochastic nature of network structural changes, the temporal views employ grouped bar charts to visualize the quantity of added and deleted edges at each time step. Additionally, area chart is embedded beneath the bar chart to represent the scale of nodes within the network. Through the temporal views, users can intuitively perceive the network's size and the change degrees at various time points. As shown in Figure 8, the timeline exhibits evident periodicity, with one week constituting a cycle. The number of nodes in the network varies between weekdays and weekends, reflecting different scales of network activity. Furthermore, the temporal views support time setting, users click the bar chart to let the main view show the layout result of the corresponding time.

### 4.2. Interaction Design

A wide range of interactions have been implemented, including zooming, selection, and filtering. Users can click the "play" button in the bottom of the main view (Figure 8B) to stop or continue the layout animation, move the slider over the temporal views (Figure 8D) to select their preferred time span, or they can take snapshot with the toolbar on the top of the main view. With these interaction techniques, users can explore dynamic network from overview to details, and the linkage between the main view and auxiliary views allows for collaborative analysis across multiple perspectives. This facilitates the discovery of the inherent characteristics and meaningful insights within the dynamic network, enabling in-depth exploration and understanding of dynamic networks.

### 4.3. Implementations

Our visualization system is implemented as a web-based application with a client-server architecture. The server side is performed using Java. For the client side, JavaScript and Vue framework is used for building the user interface and D3 is used for drawing diagrams.

## 5. Evaluations

In this section, we compare our approach with the classic and state-of-the-art approaches to evaluate the layout stability and quality. We also illustrate the effectiveness of our visualization system with use case study and in-lab user study.

*5.1. Experiments on Layout Approach*

5.1.1. Experiment Settings

**Datasets:** In terms of dataset selection, this study used the Newcomb Fraternity [46], McFarland [47], and email-Eu [48] datasets for comparative experiments, as shown in Table 1.

**Table 1.** Dynamic network datasets.

| Network | Node Count | Avg Edge Count | Steps |
|---------|------------|----------------|-------|
| Newcomb | 17 | 40 | 15 |
| McFarland | 20 | 28 | 82 |
| email-Eu | 414 | 592 | 30 |

The Newcomb dataset captures sociometric preference rankings among 17 students in a fraternity at the University of Michigan over a 15-week period. At the end of each week, these students ranked the other 16 students in order of preference from 1 to 16, with no duplicate rankings allowed. For this study, only the relationships representing the top three rankings each week were retained as edges in the network, providing insights into the changing relationships among the students. Consequently, a set of 15 networks was generated, each averaging 17 nodes and 40 edges.

The McFarland dataset, sourced from McFarland's research on classroom interactions, documents student interactions in a classroom setting to understand the learning process. It unveils the social processes involved in constructing, maintaining, and altering classroom order. The dataset records interactions among 20 students across 82 evolutionary steps.

The email-Eu dataset comprises email data from a European company, with the senders and recipients being core members of the organization. The edges in this dataset are directed, signifying the sender (from) and receiver (to) of each email, along with the timestamp of email transmission. The number of events in this network does not vary a lot in each day, and it do not contain interval across days without any event. Considering this network is relatively stable and our layout method is not timeslicing sensitive, we choose to use a simple timeslicing technique, namely the uniformed timeslicing technique, to process the data. It is worthy to note that this processing technique may hide or lose patterns on other time scales. Further timeslicing sensitive research may refer to more effective techniques, such as the non-uniformed timeslicing [38,49]. For this study, a continuous 30-day subset of data was selected, merging and processing the daily email records into an undirected network, resulting in a set of networks spanning 30 consecutive states.

**Compared Methods:** we conducted comparative experiments using the following methods: FR algorithm, Incremental algorithm [17], and Aging algorithm [16]. The FR algorithm is the most widely applied network layout algorithm, which provides a baseline for good layout quality. The Incremental algorithm and the Aging algorithms are classical online dynamic network layout approaches. The incremental algorithm is used for incremental layout adjustments. The Aging algorithm introduces node age to limit node movement probabilities.

**Stability and Quality Metrics:** For dynamic network layout algorithms, evaluation is typically conducted from two perspectives: stability and layout quality.

In terms of stability metrics, a common approach is to measure the displacement $\delta pos$ of nodes between adjacent layouts. When the network structure is updated, a smaller average displacement in the new layout indicates that there is less node movement between adjacent layouts, signifying greater stability and better maintaining the mental map. Some researchers also use layout energy as a metric, which is derived from relationships within the layout. A lower energy value signifies lower energy within the network, indicating a better layout quality. Since layout energy depends on the forces applied to nodes, and different algorithms may use distinct energy models [17]. So, to ensure comparability, this

paper adopts the average displacement metric to assess layout stability. Additionally, the layout results for each method are provided directly to assist in the evaluation process.

For layout quality, this paper employs the following three metrics [10] for a multidimensional assessment, including edge crossing metric, shape similarity metric, and edge angle of incidence metric.

**Edge crossing metric** $M_c$**:** This metric is used to quantify the number of edge crossings in a layout, with a larger value indicating fewer edge crossings in the layout. The definition of $M_c$ is as shown in Formula (8).

$$M_c = 1 - \begin{cases} \frac{c}{c_{mx}} & c_{mx} > 0 \\ 0 & otherwise \end{cases} \tag{8}$$

where $c$ represents the number of edge intersections present in the network layout, and $c_{mx}$ is an approximate upper limit of edge crossings in the network, considering that a network with $m$ edges can have up to $m(m-1)/2$ crossings without considering their degrees. Therefore, $c_{mx}$ is defined as the difference between $m(m-1)/2$ and impossible crossings, as shown in Formula (9):

$$c_{mx} = \frac{m(m-1)}{2} - \frac{1}{2} \sum_{i=1}^{n} d(v_i)(d(v_i) - 1) \tag{9}$$

**Shape similarity metric** $M_s$**:** This metric is used to assess whether the layout accurately reflects the network's shape. A higher value indicates that the layout more faithfully represents the network's shape. It is defined as the similarity between the network and the shape network generated by its layout, as shown in Formula (10).

$$M_s = \eta(G, \mu(P)) \tag{10}$$

where $\eta$ represents the similarity function between two networks, $\mu$ is the shape network function, and $P$ represents the set of node positions in the layout $L$ corresponding to the network $G$.

**Edge angle of incidence metric** $M_a$**:** This metric's quantification criterion is to maximize the minimum angle of incidence for node edges, with a larger value indicating more equal angles of incidence for nodes. It is defined as Formula (11):

$$M_a = 1 - \frac{1}{|V|} \sum_{v \in V} |\frac{\theta(v) - \theta_{min}(v)}{\theta(v)}| \tag{11}$$

where $\theta_{min}(v)$ represents the minimum angle of incidence for the edges of node $v$, $\theta(v)$ represents the ideal angle of incidence for node $v$ , and it is defined as $360°/d(v)$ .

### 5.1.2. Layout Result Analysis

Firstly, we present layout results of the three datasets using different algorithms.

**Analysis on The McFarland Dataset:** For the McFarland dataset, Figure 10 illustrate the layout results for five consecutive states under different algorithms.

From the above layout result, all algorithms except FR can maintain structural stability in minor changes (from time step 1 to 3); Our algorithm better preserve the relative position on significant change (step 4) and on adding back a node (step 5). The detailed analysis is illustrated as follows.

For the first three states, the network structure undergoes small changes, all algorithms maintain the shape of this structure well except for the FR algorithm.

When transitioning from the third state to the fourth state, there is a significant transformation in the network structure, with most nodes and edges disappearing, leaving only a substructure consisting of nodes 4, 5, 7, 10, and 12 (highlighted in the figures). Our algorithm's layout better preserved the relative positions of nodes. This is because

our algorithm considers the structural-based influence on different nodes, the node 7 has a greater degree of freedom, while the other nodes is reduced, resulting in smaller positional changes.

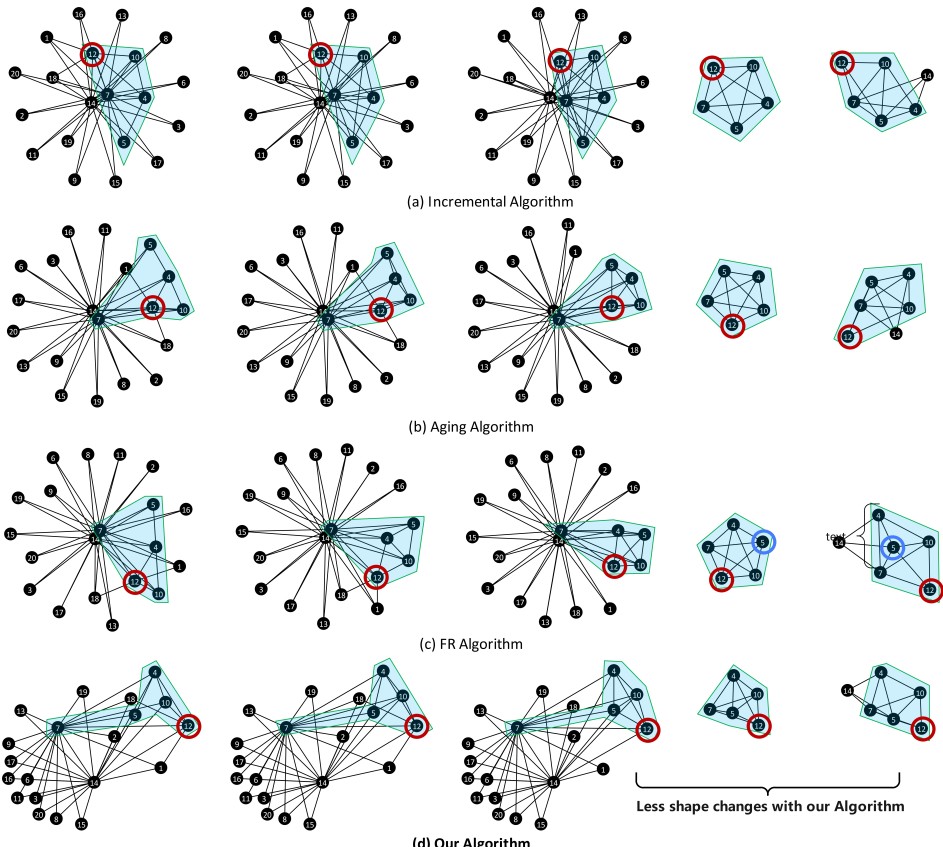

**Figure 10.** Layout result comparison based on McFarland data. Node 12 and node 5 are highlighted with red and blue circles, their spatial and relative positions are kept stable in our algorithm and the incremental algorithm, but changes in the other two algorithms. The shape of the blue-color-highlighted structure is better preserved with our algorithm on significant change (step 4) and on adding back a node (step 5).

When the network updates from the fourth state to the fifth state and adds back previously deleted node 14, the shape of this structure undergoes significant changes in the layout generated by all algorithms except for our algorithm.

**Analysis on The Newcomb Dataset:** For the Newcomb dataset, Figure 11 displays the layout results for the first ten consecutive states under different algorithms. Unlike the McFarland dataset, the Newcomb dataset maintains a consistent number of nodes across all states, and the variations in the network structure are solely attributed to changes in edges.

From the above layout result, the Incremental algorithm exhibits better stability than the other three algorithms across all time frames though there are substantial changes in actual network; Only the Incremental algorithm and our algorithm preserve the relative positions of nodes within the structure for some structure changes (as the example highlighted portions in states 6–9). We elaborate the detailed analysis as follows.

First, when considering the overall layout shapes, the Incremental algorithm exhibits relatively consistent layout shapes across all time frames, while the layouts generated by the other algorithms undergo more significant changes. However, it's important to note that the actual network structure experiences substantial changes itself, and the Incremental algorithm excessively restricts node movement, resulting in layouts that do not effectively reflect the network's structural changes.

Second, for some structures with minor changes, such as the highlighted portions in states 6–9, only the Incremental algorithm and our algorithm preserve the structure shape and the relative positions of nodes within the structure.

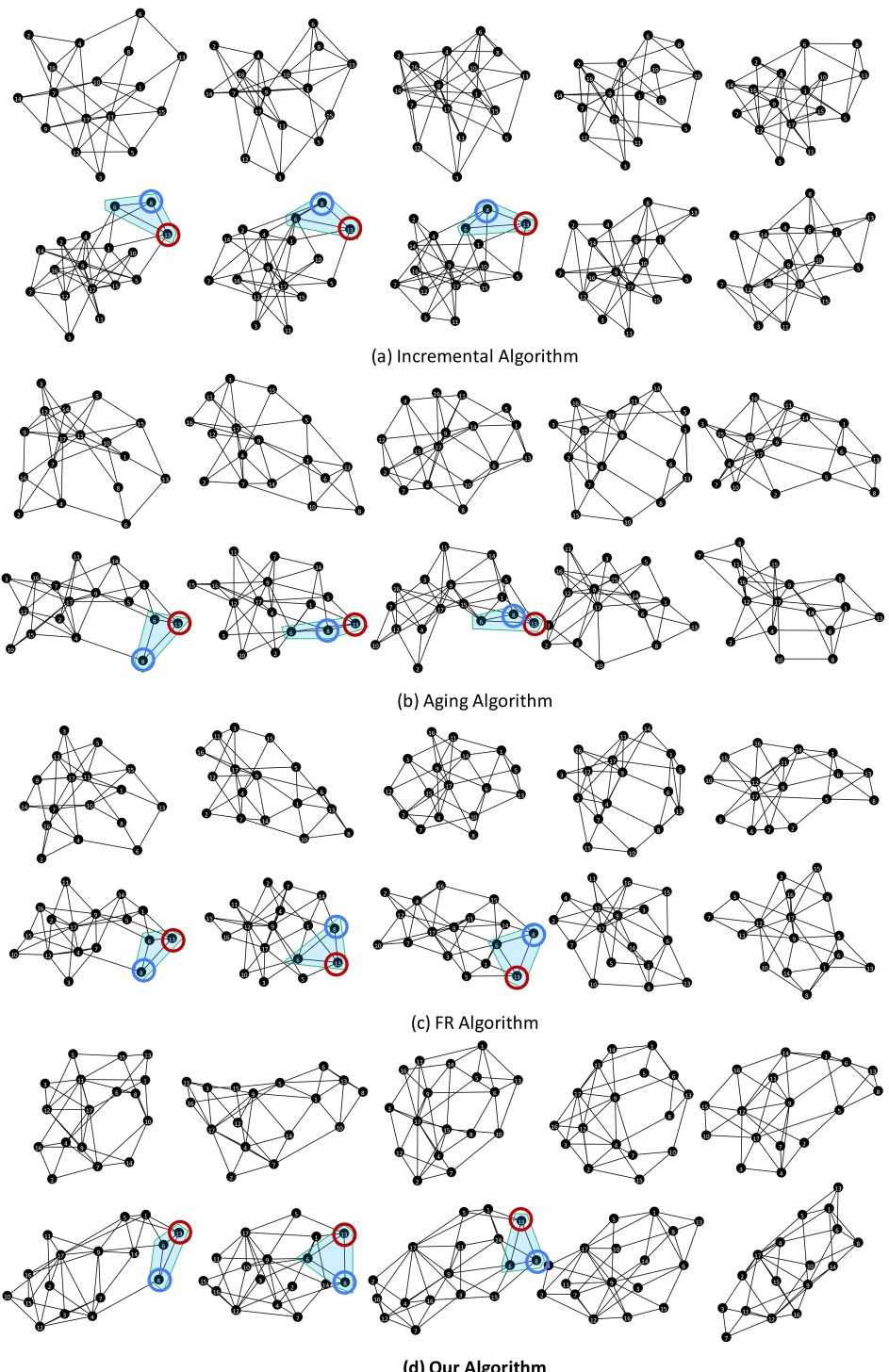

**Figure 11.** Layout result comparison based on the Newcomb data. Node 13 and node 8 are highlighted with red and blue circles, their spatial and relative positions are kept stable in our algorithm and the incremental algorithm, but changes in the other two algorithms.

**Analysis on The Email-Eu Dataset:** For the emai-Eu dataset, which is large in scale and exhibits significant changes in network structure between adjacent timeframes, it is challenging to directly compare and observe layout differences at different timeframes.

Therefore, this paper selected a specific day's network structure with a two-hour time step to showcase the formation process of this structure as the fourth experimental dataset. As shown in Figure 12, the layout results for timeframes 3 to 6 are displayed.

From the layout results, the Incremental algorithm present least position changes, but due to certain node positions are overly fixed, we can see the presence of long edges and excessively clustered nodes. In comparison, our algorithm retain the shape and relative positions of some unchanged structures, while also present the overall shape of the network.

Based on the layout results on these three datasets, our algorithm achieves a balance between layout stability and quality. It better preserves structural and relative position on significant changes (on time step 4–5 of the first dataset), yet it gives space for reflecting the actual network changes (on the second dataset) and avoiding extreme layout such as too clustered nodes or long edges (on the third dataset).

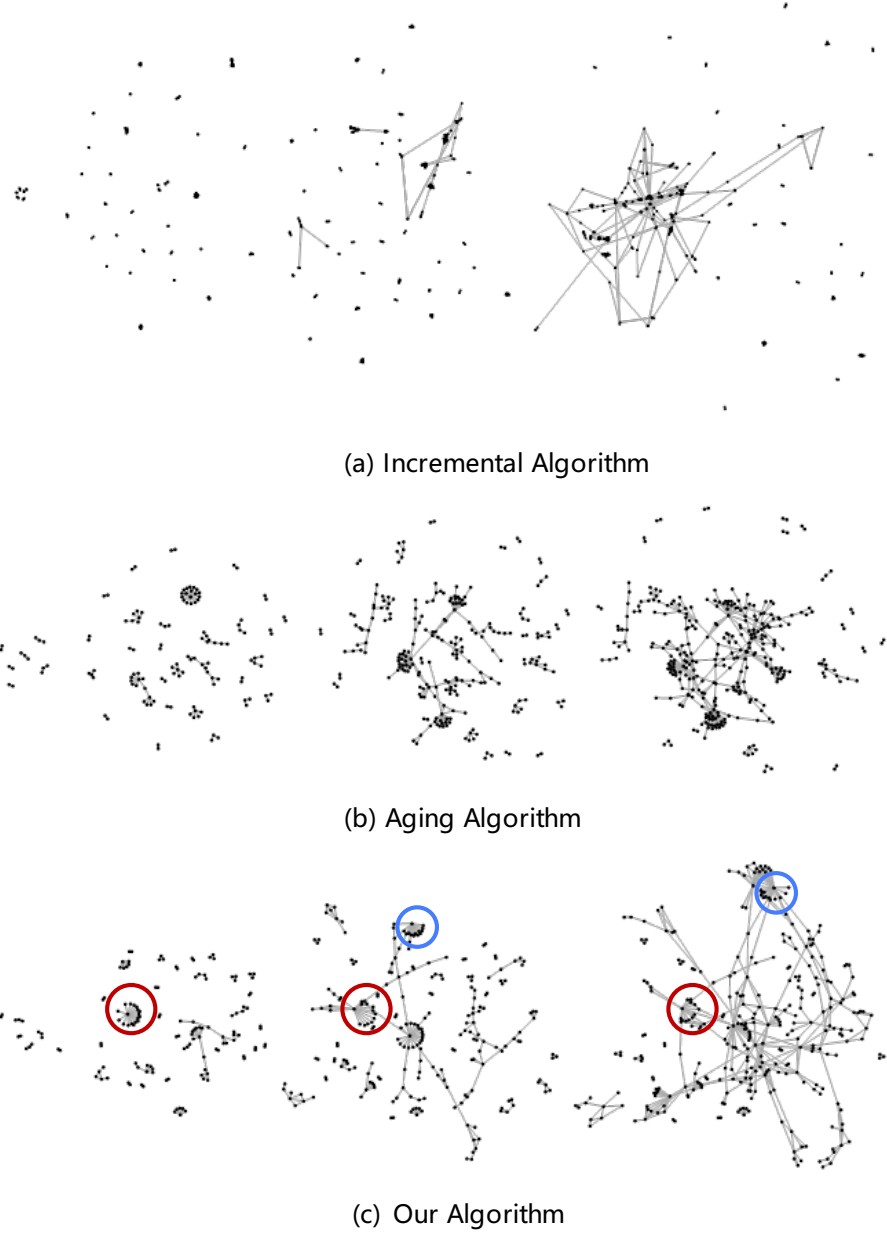

(a) Incremental Algorithm

(b) Aging Algorithm

(c) Our Algorithm

**Figure 12.** The layout result comparison based on the email-Eu_day1 data. Stable structures highlighted with red and blue circles can be found in the layout by our algorithm, but not in the layouts by the other algorithms. Long edges and excessively clustered nodes exists in the layout by Incremental algorithm.

### 5.1.3. Quantitative Evaluation and Results Analysis

**Result of network stability:** We computed statistics on the average displacements of nodes between adjacent layouts for different algorithms on three datasets to measure degrees of node movement between adjacent layouts. The results are presented in Table 2.

**Table 2.** Average displacements. The best value for each metric is highlighted in bold.

| Network | Aging | FR | Incremental | Ours |
|---------|-------|-----|-------------|------|
| Newcomb | 2.6905 | 3.9169 | **1.0798** | 3.1675 |
| McFarland | 0.3969 | 1.0571 | 0.9594 | **0.3851** |
| email-Eu | 9.4554 | 15.1272 | **9.1390** | 12.1970 |
| email-Eu_day1 | 0.84235 | 6.1739 | **0.6715** | 4.5652 |

As seen from the table, for average displacement, our algorithm outperforms FR algorithm, similar to Aging algorithm, and not as good as Incremental algorithm. Except for the McFarland dataset, the Incremental algorithm consistently achieves the smallest average displacement, aligning with the previous layout result analysis. The average displacement of our algorithm is larger than that of Incremental algorithm. This is because our algorithm makes some trade-offs on the impact of changes in network structure on the entire network.

**Result of layout quality:** Finally, we assessed the layout quality of different algorithms. The results are presented in Tables 3–5 below.

**Table 3.** Edge crossing metric. The best value for each metric is highlighted in bold.

| Network | Aging | FR | Incremental | Ours |
|---------|-------|-----|-------------|------|
| Newcomb | 0.9361 | 0.9394 | 0.8979 | **0.9468** |
| McFarland | 0.9236 | 0.9292 | 0.8910 | **0.9430** |
| email-Eu | 0.9949 | 0.9962 | 0.9916 | **0.9968** |
| email-Eu_day1 | 0.9948 | **0.9970** | 0.9953 | 0.9952 |

**Table 4.** Edge angle of incidence metric. The best value for each metric is highlighted in bold.

| Network | Aging | FR | Incremental | Ours |
|---------|-------|-----|-------------|------|
| Newcomb | 0.3388 | 0.3334 | 0.2860 | **0.3723** |
| McFarland | 0.5710 | **0.6498** | 0.5506 | 0.6040 |
| email-Eu | 0.7733 | 0.7725 | 0.7743 | **0.7848** |
| email-Eu_day1 | 0.81342 | 0.8203 | **0.8496** | 0.8165 |

Based on the above layout quality metrics, except the email-Eu_day1 dataset, our algorithm produces layouts with the smallest number of edge crossings, and our algorithm also outperforms or closely match the FR algorithm in terms of edge angle of incidence and shape similarity, indicating that our algorithm achieves better results in network layout quality. Incremental algorithm achieves best layout quality on the email-Eu_day1 dataset, highlighting its effectiveness for incremental data like email-Eu_day1.

**Table 5.** Shape similarity metric. The best value for each metric is highlighted in bold.

| Network | Aging | FR | Incremental | Ours |
|---------|-------|-----|-------------|------|
| Newcomb | 0.3362 | **0.3812** | 0.2905 | 0.3515 |
| McFarland | 0.4292 | 0.2723 | 0.3526 | **0.4854** |
| email-Eu | 0.2289 | 0.2500 | 0.2838 | **0.3090** |
| email-Eu_day1 | 0.2928 | 0.3371 | **0.3916** | 0.3368 |

In summary, from both the layout result analysis in Section 5.1.2 and the quantitative metric analysis in Section 5.1.3, the SIPA algorithm achieves a balance between network stability and layout quality. Though the SIPA produces slightly larger node displacement between adjacent layouts than that of the Incremental and Aging algorithms, it brings chances for more nodes to seek ideal positions globally, thus better preserving overall structural shapes and relative positions within structures on significant network changes, and it presents better layout quality comparing to the other two algorithms with constraints.

### 5.2. Informal Evaluation on Visualization System

To demonstrate the effectiveness and usability of our visualization system, we conduct evaluations through case studies and user experiments.

### 5.2.1. Use Cases

The email-Eu dataset was chosen for case studies, aiming to analyze and explore the dataset to validate the utility of the system. First, there is the variation in network scale. Based on the temporal views (Figure 13), the network scale exhibits clear periodicity, with a cycle of 7 days aligning with the patterns of weekdays, weekends, and a three-day holiday in the second week. Additionally, there are subtle differences in the network scale during weekdays, with Monday and Friday having slightly larger networks than the other workdays. This aligns with people's work habits, as emails are typically sent before and after weekends. Furthermore, the temporal views reveal significant structural changes as the network evolves, with a substantial number of edges being added and removed between adjacent networks.

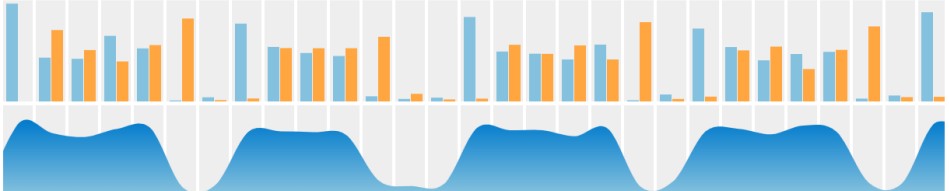

**Figure 13.** Temporal views with periodicity.

Secondly, there are similar structures within the email network. While observing the evolution of the network, the network contains a significant number of "star-shaped structures", as shown in Figure 14. In the context of real data, nodes in the network represent email senders or recipients. The reason for the frequent occurrence of these "star-shaped structures" is attributed to mass email distribution. Senders employ the group email feature to send emails to multiple recipients simultaneously, but there are no email records between the recipients, leading to this start network structure.

Lastly, for adjacent network structures comparison: As indicated by the temporal views, the structural changes between adjacent networks are quite substantial, making it challenging to directly compare their structural differences. The Figure 15 presents snapshots of the network at the 5th and 6th time steps. The upper part directly shows the layout results, three structures seem to be retained after the network update (marked with blue circles), concerning relative positions and structural shapes. However, when we use 3-step animation to highlight different node changes (as shown in the bottom of Figure 15), emphasizing deleted, newly added, and retained nodes and structures. It becomes apparent that, despite the top right and the bottom structures (marked with blue circles in the top part) having similar shapes and relative positions as the former time step, they are newly added structures. Only top left structure is retained.

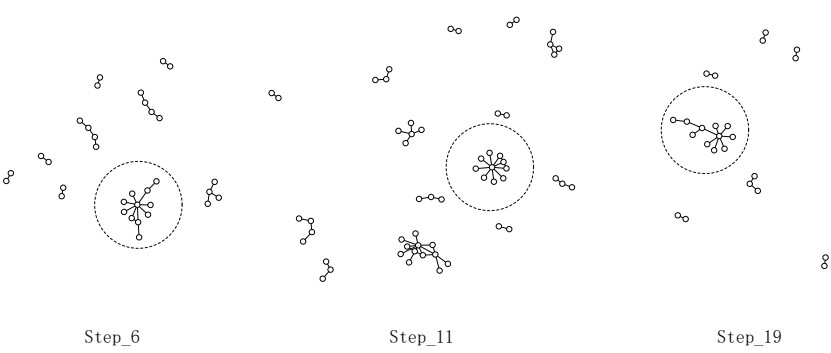

Step_6            Step_11            Step_19

**Figure 14.** Similar star-shaped network structures.

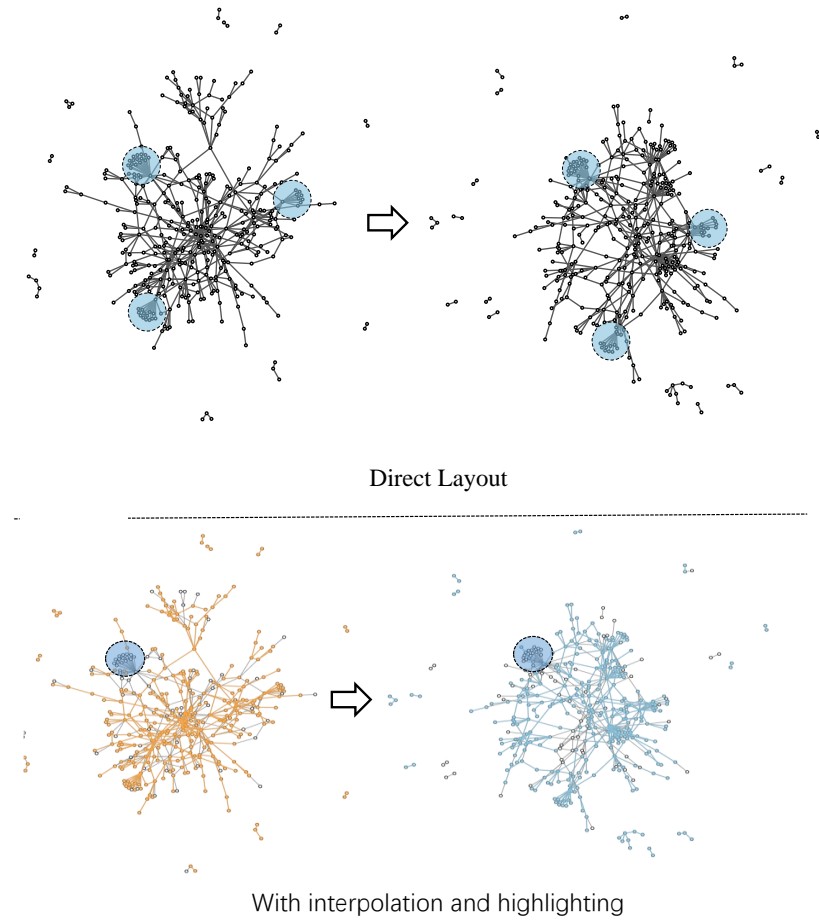

**Figure 15.** The network structure comparison for time step 5 (**left**) and 6 (**right**).

### 5.2.2. User Experiment

To demonstrate the usability and utility of our system, we conducted a system evaluation with recruited users. First, below four tasks are designed based on the three datasets:

1.  Identify changes in network scale (increase, decrease, or unchanged).
2.  Determine if there have been significant changes in network structure.
3.  Identify whether specific structures have been preserved.
4.  Provide the previous layout position of specific nodes.

Then, 15 volunteers were recruited, including 10 undergraduate students and 5 graduate students. We first demonstrated our system to them, and then they were asked to

complete the four tasks using our system. After the volunteers completed the tasks, they were invited to rate the system for its usability, readability, interpretability, and effectiveness on a scale ranging from 0 (strongly disagree) to 5 (strongly agree). These ratings were used to evaluate the system's performance in assisting users in completing the tasks.

The statistical results, as shown in Figure 16, indicate that the users generally had a high acceptance level for the visualization system. The system are helpful in completing routine analysis tasks and enabling users to understand and explore the data in a straightforward manner. Through the system's interactive operations and highlighting features, users could rapidly grasp changes in the network's structure, facilitating exploration of the network dynamics.

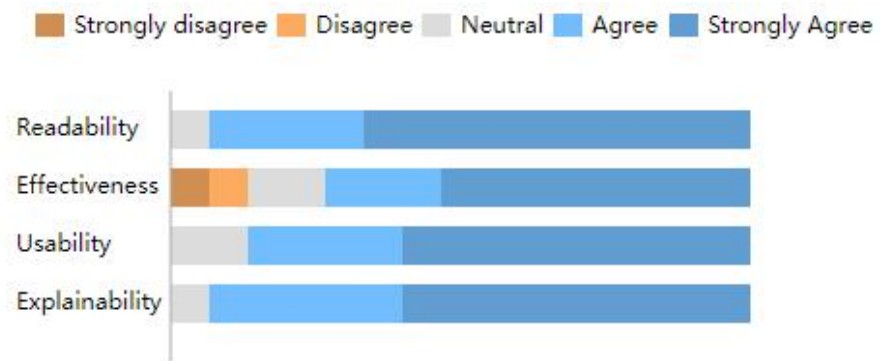

**Figure 16.** The user study result based on Likert scale.

While the visualization system is helpful to users in understanding network changes and capturing relatively stable structures, there were instances where the effectiveness of the system received low ratings from two volunteers, scoring 0 and 1, respectively. Through user interviews, we found that these two users struggled when dealing with large-scale dataset (the email dataset) or when the network underwent significant structure changes. In such scenarios, users found it challenging to track changes, especially for multiple structural changes within the network, resulting in lower effectiveness ratings. This inspired us a direction for future optimization efforts, where we will explore potential technologies to enhance the comparative analysis of large-scale networks and the multi-focus tracking.

## 6. Discussions

Our method focuses on preserving stable structures in the node-link-diagram-based online dynamic network visualization. Comparative experiments demonstrate that our method better maintains the shape and relative node positions of stable structures as the network evolves. These stable structures provide anchors for users' tracing of network evolvement. The general feedback from user study is positive; however, the feedback on larger network (the email network data [48], with hundreds of nodes and edges) is not promising. We would like to discuss the scalability limitation in this section.

Although scalability has been discussed in-depth for static graph, it has only played a minor role in designing most dynamic graph visualization approaches , from Fabian Beck's dynamic visualization survey [34] in 2014. In recent years, several more scalable techniques have been proposed. The layout form of massive sequence view [37,38] provides consistent node positions across the entire time, and it is able to detect patterns in relatively larger network (150 nodes is presented in [38], networks within 1000 nodes are tested in [37]). Various time-slicing techniques [38,49] are proposed to optimized the scalability on time dimension. Also network simplification techniques such as filtering [6], aggregating [50–53] are proposed to reduce the large network to manageable size. But due to the spatial sparseness nature of node-link diagram [54], direct presenting a large-scale dynamic network with node-link diagram is still a challenging problem. In the following paragraphs, we discuss our scalability limitation from algorithmic, visual, and perception

aspects per Richer's scalability model [55]. We also discuss workarounds for this limitation. For visualizing large-scale networks, the joint employment of interaction techniques, or network simplification techniques may be required.

**Algorithm Scalability:** our method comprises of three steps executed sequentially. Supposing a network with $|V|$ vertices, $|E|$ edges, and $|delta|$ node changes on average, the most complex step costs $O(|delta| * (|V| + |E|))$ in time and $O(|V|)$ in space. The running time of all three test cases are within dozens of milliseconds. The complexity is at a similar level to existing works [14,17]. The algorithmic scalability is not a major issue in our case.

**Visual Scalability:** Visual scalability primarily maps to readability [55]. Our method is built upon the node-link diagram layout. The node-link representation by design leaves significant background space empty and thereby may encounter scalability problems when applied to larger graphs [54]. Thus it is not as scalable as matrices, space-filling layouts, or the massive sequence view. Although our method improves the node-link layout quality by preserving stable structures, its readability on a large-scale dynamic network is still challenging. In practice, node-link diagrams are extremely common and widely employed [15], and are suitable to present the global overview and high-level changes [16] and perform better on path tracing tasks [36]. So there are usage scenarios of visualizing large node-link diagram. For these cases, visual abstraction interactions, including highlighting, brushing, and semantic zooming as in an overview + details-on-demand approach, may be required to enhance the visual scalability [1].

**Perceptual and Cognitive Scalability:** Perception and cognitive scalability investigate the scalability of human perception and cognition when performing a visualization task [55]. Evaluating the cognitive load on the user during the insight generation process is a challenge that crosscuts general, large, and dynamic network visualization [55]. We evaluated user perception on our system through user studies, the general feedback is positive. Yet more rigorous evaluation is expected in the future, such as the controlled user experiment comparing the perception on the node-link and the matrices diagrams [36], or the animation versus static display [33]. As a complementary, we also provide several designs to enhance user perception, adopting highlighting, zooming and time slider interactions, and providing multiple views.

**Workarounds:** Before discussing workarounds, we would like to make clarification on the definition of "large" in our context. There are different definitions for "large-scale", ranging from 50 nodes [56] to thousands of nodes [57] and even to millions of nodes [58]. These variations account for differences in the domain, the data itself, tasks and visual techniques [56]. In the context of our presentation, directly presenting the network with node-link diagram and considering user perception capabilities, the number is relatively small. In the empirical study of literature on human-centered experiments [56], networks with 51 to 200 nodes are identified as a large network. According to this identification and combining our user experiment results, we define networks with more than 50 nodes as a large network in our context. To address the issue of visual clarity with large networks (with more than 50 nodes), other than using more sophisticated layout techniques, algorithms for network simplification have been investigated [1]. Network simplification techniques such as aggregating [50–53], sampling [21,39,59–62], and filtering [6] minimize the problem scale while keep important information. As a workaround, for a large-scale network, users can first use these simplification techniques to reduce the network to a manageable size, then use our method to compute the layout of the final network.

In light of the above discussions, we propose that our method can support the large-scale online dynamic network visualization in two ways. First, for usage scenarios of directly presenting large-scale dynamic networks with node-link diagrams, our method can better preserve stable structures of the network, while the challenge on visual and perception efficiency can be alleviated by interaction approaches such as highlighting, panning-and-zooming, and filtering; or it can be used together with other visual forms in multiple-view designs, as a "level-of-detail" or an "overview + details-on-demand" approach. Such example design can be found in egoSlider [6] which presents large dy-

namic network in node-link diagrams with multi-focus-highlighting and with multiple coordinated views. Second, for the scenarios where network simplification techniques are used, our method can act as a visual effect for the final layout. The clustering, sampling, bundling, and filtering techniques acquire representative "small networks" of the original network. Their analysis processes generally involve observing the "small network". For small network visualization, node-link diagrams are more readable and more familiar than matrices [36]. Our method can then be used to present the "small network" evolution in a coherent manner.

### 7. Conclusions and Future Work

This paper introduces the Structural Influence and Preservation (SIPA) Algorithm as a solution to the challenge of preserving stable node structures in online dynamic network visualization. The SIPA employs a novel strategy to compute and propagate node influence within the network based on structural changes and incorporates a node aging mechanism. The algorithm's performance is evaluated through layout result analysis and quantitative metrics. Results show that SIPA can better maintain the relative node positions and shapes of stable structures, thereby enhancing the overall layout stability. In addition, given that comprehending network dynamics solely through network layout results is challenging, we complement the algorithm with an interactive visualization system, enriching the layout results with various interactions and views of temporal context, network features, animated graph-diaries and network snapshots. Case studies and user experiments demonstrate the system's effectiveness in analyzing and tracking network changes over time.

In future work, we would like to study the possible techniques to extend our research to address the challenges of visualizing large-scale networks. We will adopt multilevel approaches and community detection algorithms to partition extensive networks into manageable layers, allowing users to analyze network dynamics at various levels. Additionally, we plan to develop adaptive algorithms to automatically adjust the multiple parameters within our current algorithm, enhancing its adaptability and efficiency. These efforts will make large-scale dynamic network analysis more accessible, expanding the applicability of our visualization framework in the process.

**Author Contributions:** Conceptualization, G.W. and Y.W.; methodology, G.W. and H.C.; software, R.Z., H.C., J.L. and W.G.; validation, H.C., W.G. and F.W.; formal analysis, G.W. and R.Z.; investigation, G.W.; resources, Y.W.; data curation, R.Z.; writing—original draft preparation, G.W. and H.C.; writing—review and editing, G.W.; visualization, G.W.; supervision, Y.W.; project administration, G.W.; funding acquisition, Y.W. All authors have read and agreed to the published version of the manuscript.

**Funding:** This research was funded by the National Natural Science Foundation of China with grant number 61872304, and the Talent Project of Sichuan University of Science and Engineering with grant number 2020RC20.

**Institutional Review Board Statement:** Not applicable.

**Informed Consent Statement:** Not applicable.

**Data Availability Statement:** Data available in a publicly accessible repository. The data presented in this study are openly available in https://snap.stanford.edu/data/.

**Conflicts of Interest:** The authors declare no conflict of interest. The funders had no role in the design of the study; in the collection, analyses, or interpretation of data; in the writing of the manuscript; or in the decision to publish the results.

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
