# Peer review of "Online Dynamic Network Visualization Based on SIPA Layout Algorithm"

_applsci, doi:10.3390/app132312873_

Round 1

Reviewer 1 Report

Comments and Suggestions for Authors

This paper presents a node organization method for animated node-link diagrams focused on online temporal networks. The authors’ algorithm is compared with different node organizations using three distinct networks. They conducted a qualitative and quantitative evaluation analyzing the network stability and quality under four different metrics. A user study was also conducted with 15 participants to evaluate the usability and utility of their solution. Although the paper does a good job of evaluating their solution through all those different approaches (quantitative and qualitative), many improvements need to be addressed, and there are several missing discussions.

The paper needs to be improved into four different aspects: in relation to (1) terminology, when compared with state-of-the-art literature, (2) related works, the paper is missing discussions in a number of important related papers, (3) the case study and user study needs to be improved to be more convincing, (4) the authors should discuss and position their approach in relation to scalability, and (5) the authors should follow a more formal writing style. Each point is discussed in more details below.

(1) Terminology:

·         Node-link diagram: the authors should use better the terminology in relation to the name of the used layout.

·         Network structure: The authors used the term network structure but did not previously state or compare what where they intention. For instance, to network structure, you have two perspectives: the structure in each timestamp, for instance, if that node-link represents a tree, star, circle, and so on (see [1]), or between timestamps, such as the lifecycle of those structures between timestamps, if the nodes died, appeared, merged with other nodes, and so on (see [2]). This terminology should be discussed and used over the paper when demonstrating some related concept.

[1] C. Li, G. Baciu, and Y. Wang. Module-based visualization of large-scale

graph network data. Journal of Visualization, 20(2):205–215, May 2017

[2] L. R. Pereira, R. J. Lopes, and J. Louca. Community identity in a temporal

network: A taxonomy proposal. Ecological Complexity, 45:100904, 2021.

·         Online and streaming: the related literature often use the term online and stream for the same or similar purposes. The authors should point out that and discuss it in the related work.

·         Communities: the authors introduced in the middle of the paper the concept of communities, but barely discuss anything about it. This term should be introduced, and better discussed what is meant in the paper. For instance, how were these communities calculated? Do the authors use any of the popular algorithms for community detection (e.g., Infomap or Louvain)? These discussions should be included in the paper.

·         Timestamp and timeslicing: during the experiments, the authors decided to aggregate the timestamps into a small defined number of "steps" to simplify the analysis. However, how this number was choosen? How can they guarantee that there is no information hidden or lost in this number of steps? The literature usually addresses this problem as uniform and nonuniform timeslicing. In my understanding, the authors decided to use a uniform approach. However, the choice of this approach has been criticized by the literature [3]. This should be included in the paper and justified.

[3] Ponciano J (2021). An online and nonuniform timeslicing method for network visualisation. Computers & Graphics

(2) Related work:

Besides the related works mentioned in the terminology discussion, the paper still misses a number of papers that have been proposed to analyze and explore (online and offline) temporal networks. For instance, at the end the authors are using an animated node-link diagram to visualize and explore online temporal networks. However, there are several complementary approaches in recent literature that study different visualizations for the same problem. Since there are few or no discussions related to this aspect, this should be greatly improved in sections 2.1 and 2.2.

For instance, the following questions should be considered and answered: why animated node-link diagrams over other techniques [4]? why not small multiples? why not Massive Sequence View [5]? matrix-based layouts?

[4] Filipov, V., Arleo, A. and Miksch, S. (2023), Are We There Yet? A Roadmap of Network Visualization from Surveys to Task Taxonomies. Computer Graphics Forum

[5] S. v. d. Elzen, D. Holten, J. Blaas and J. J. van Wijk, "Dynamic Network Visualization with Extended Massive Sequence Views," in IEEE Transactions on Visualization and Computer Graphics, vol. 20, no. 8, pp. 1087-1099, Aug. 2014, doi: 10.1109/TVCG.2013.263.

(3) Case study and user study:

The results presented in the case study are not too convincing. For instance, Figures 8, 9, and 10 demonstrate their solution when compared with similar state-of-the-art approaches, but it is very hard to see what the upgrades are compared to the other similar approaches.

The authors considered the aging aspect of the animation. How the speed of the animation was evaluated?

Also, the authors mention that they are using a hybrid approach, but for the case and user studies sections, it is hard to understand the importance of the hybrid aspect of the solution.

At last, the user study results (Figure 12) demonstrate findings related to the structure of the Enron network. However, since it does not ask the users to compare with different node organization approaches, how can the users know that the authors' approach is better than the other ones?

(4) Scalability:

The paper is missing a complete discussion about scalability, which, for me, is the most important issue of the paper. The networks tested were fairly small in the number of nodes, edges, and timestamps. The results of the largest network (Enron) are very hard to understand and to find any pattern at all (Figure 10). The paper should be positioned in relation to scalability in the field [6]. Also, papers addressing a similar problem of analyzing online and offline temporal networks usually use sampling techniques [6-13], timesliscing approach [14-15], and aggregating solutions [16-19], specifically made because of this scalability issue. Since the paper is missing a complete discussion on this point, I am not sure why I should select the authors' solutions over the existing ones, that deal so much better with larger networks.

[6] G. Richer, A. Pister, M. Abdelaal, J. -D. Fekete, M. Sedlmair and D. Weiskopf, "Scalability in Visualization," in IEEE Transactions on Visualization and Computer Graphics, doi: 10.1109/TVCG.2022.3231230.

[7] Ahmed NK , Neville J , Kompella R . Network sampling: from static to streaming graphs. ACM Trans Knowl Discov Data 2013;8(2):7:1–7:56 10.1145/2601438 .

[8] Ahmed NK, Duffield N, Willke TL, Rossi RA. On sampling from mas- sive graph streams. Proc VLDB Endow 2017;10(11):1430-1441. doi: 10.14778/ 3137628.3137651

[9] Etemadi R, Lu J. PES: Priority edge sampling in streaming triangle estimation. IEEE Trans Big Data 2019:1. doi: 10.1109/TBDATA.2019.2948613

[10] Zhao Y, Chen W, She Y, Wu Q, Peng Y, Fan X. Visualizing dynamic network via sampled massive sequence view. In: Proceedings of the 12th international symposium on visual information communication and interaction, VINCI’2019. New York, NY, USA: ACM; 2019. p. 32:1–32:2. doi: 10.1145/3356422.3356454 . ISBN 978-1-4503-7626-6

[11] Ponciano, J., et al (2021). A streaming edge sampling method for network visualization. Knowledge Information System

[12] Sarmento R, Cordeiro M, Gama Ja. Streaming networks sampling using top- k networks. In: Proceedings of the 17th international conference on enter- prise information systems –Volume 1, ICEIS 2015. Setubal, PRT: SCITEPRESS –Science and Technology Publications, Lda; 2015. p. 228-234. doi: 10.5220/ 0 0 05341402280234 . ISBN9789897580963

[13] Ahmed NK, Neville J, Kompella R (2013) Network sampling: from static to streaming graphs. ACM Trans Knowl Discov Data 8(2):7:1–7:56. https://doi.org/10.1145/2601438

[14] Wang Y , Archambault D , Haleem H , MoellerT , Wu Y , Qu H .Nonuniform timeslicing of dynamic graphs based on visual complexity. In: Proceedings of the 2019 IEEE visualization conference (VIS). IEEE; 2019. p. 1–5 .

[15] Ponciano J., Linhares C., Faria E., Travençolo B. (2021). An online and nonuniform timeslicing method for network visualisation. Computers & Graphics

[16] Sikdar S, Chakraborty T, Sarkar S, Ganguly N, Mukherjee A (2018) Compas: Community preserving sampling for streaming graphs. In: Proceedings of the 17th international conference on autonomous agents and multiagent systems, pp 184–192. International foundation for autonomous agents and multiagent systems

[17] Stanley, N., Kwitt, R., Niethammer, M. et al. (2018). Compressing Networks with Super Nodes. Scientific Reports

[18] Kruiger, J., Rauber, P., Martins, R., Kerren, A., Kobourov, S., Telea, A. (2017). Graph Layouts by t-SNE. Computer Graphics Forum

[19] S. van den Elzen, D. Holten, J. Blaas and J. J. van Wijk, "Reducing Snapshots to Points: A Visual Analytics Approach to Dynamic Network Exploration," in IEEE Transactions on Visualization and Computer Graphics, vol. 22, no. 1, pp. 1-10, 31 Jan. 2016, doi: 10.1109/TVCG.2015.2468078.

(5) Writing formalism:

The paper was structured using several topics, bullet points, numbers to divide paragraphs, and so on, which I think should be reconsidered to be more formal. For instance, in section 5.1.3. the authors presented (1) Result of network stability. And (2) the Result of layout quality, which is very unformal.

Overall, I recommend the paper go under a major revision considering all the aspects mentioned above.

Author Response

Dear Reviewer,

Thank you sincerely for your valuable comments and suggestions regarding our manuscript.

Please see the attachments for our reply. Only one attachment can be uploaded, so would you please find the revised manuscript, and the revised manuscript with highlighting from the editors?

We look forward to hearing your thoughts on the revised manuscript. Please kindly let us know if any further information is needed from us.

Sincerely

Guijuan

Reviewer 2 Report

Comments and Suggestions for Authors

The authors should take my comments into account and address the necessary corrections listed below.

1) Online Dynamic Network Visualization is a very important and hot topic. Authors should add and emphasize the motivation part of this study.

2) Although it is stated that the study contains an important application, the stages of the application and the flow diagram are not followable. Evaluation of the results is insufficient. The first 4 chapters contain book information. Afterwards, direct results are given.

3) It should be presented especially supported by infographics.

4) Which software tool is Figure 6? Graphdiaries?

5) Academic richness in study should be improved.

Comments on the Quality of English Language

it can be improve.

Author Response

(The authors gave the same response as above.)

Reviewer 3 Report

Comments and Suggestions for Authors

The paper is well-written. The main contribution, research gap, research innovation, and novelty are clearly defined and comprehensive. 

My main concern about the paper is that the benefit to the reader is not obvious. The purpose is unclear. Is it a tutorial? If it is, there are no directions for the reproducibility of the algorithm proposed and no access to the visualization platform for a potential user. Many details, like required data structure and data loading, are missing.

The authors should summarize their methodology in an algorithm and a flowchart so readers can follow it more easily. Thus, the practitioners could have some guidelines when applying this methodology.

Author Response

(The authors gave the same response as above.)

Round 2

Reviewer 1 Report

Comments and Suggestions for Authors

I think that the authors did a good job including my comments and considerations on the main paper, which now represents much better the terminology used in the visualization field, and better covers the related papers. Also, now the contributions and results are much better presented and convincing in relation to comparable techniques. At last, the paper discusses and positions itself to one of the main problems in the visualization field, which is scalability.

For a minor review, since now the paper includes an important discussion about small/large networks, I would suggest that the authors follow how recent papers have been defining and creating their specific definitions for the size of networks. For instance, studies considering non-temporal networks without working on visualization define large networks as millions of nodes [1]. However, a survey comparing different-sized networks for visualization considers networks with more than 200 nodes as very large [2]. More recently, the study for temporal network visualization [3] considered large temporal networks as those with a few thousand nodes or timestamps.

[1] P. Mi, M. Sun, M. Masiane, Y. Cao, and C. North. Interactive graph layout of a million nodes. Informatics, 3(4), 2016.

[2] V. Yoghourdjian, D. Archambault, S. Diehl, T. Dwyer, K. Klein, H. C. Purchase, and H.-Y. Wu. Exploring the limits of complexity: A survey of empirical studies on graph visualisation. Visual Informatics, 2(4):264 –282, 2018.

[3] C. D. G. Linhares, J. R. Ponciano, D. S. Pedro, L. E. C. Rocha, A. J. M. Traina and J. Poco, "LargeNetVis: Visual Exploration of Large Temporal Networks Based on Community Taxonomies," in IEEE Transactions on Visualization and Computer Graphics, vol. 29, no. 1, pp. 203-213, Jan. 2023, doi: 10.1109/TVCG.2022.3209477.

Author Response

Dear Reviewer,

Please see the attachment for the detail.

Best Regards,

Guijuan Wang

Reviewer 2 Report

Comments and Suggestions for Authors

I congratulate the authors for their work on such new and current topics.

I recommend that the author read the new and updated Graf articles that I published in 2023 and, if possible, enrich the references by adding them to the literature.

[1 M. Soylu, A. Soylu, ve R. Das, “A new approach to recognizing the use of attitude markers by authors of academic journal articles”, Expert Systems with Applications, c. 230, sy 120538, Kas. 2023, doi: 10.1016/j.eswa.2023.120538.   [2] R. Das ve M. Soylu, “A key review on graph data science: The power of graphs in scientific studies”, Chemometrics and Intelligent Laboratory Systems, c. 240, sy 104896, Haz. 2023, doi: 10.1016/j.chemolab.2023.104896.   Comments on the Quality of English Language

It can be improved. 

Reviewer 3 Report

Comments and Suggestions for Authors

The revisions are well done. The authors addressed all my comments adequately.
